# Bidirectionally promoting assembly order for ultrastiff and highly thermally conductive graphene fibres

Peng Li[1,5], Ziqiu Wang[1,5], Yuxiang Qi[1,5], Gangfeng Cai[1], Yingjie Zhao[2], Xin Ming[1], Zizhen Lin[3], Weigang Ma [3], Jiahao Lin[1], Hang Li[1], Kai Shen[1], Yingjun Liu [1,4] ✉, Zhen Xu [1,4] ✉, Zhiping Xu[2] ✉ & Chao Gao [1,4] ✉

Macroscopic fibres assembled from two-dimensional (2D) nanosheets are new and impressing type of fibre materials besides those from one-dimensional (1D) polymers, such as graphene fibres. However, the preparation and property-enhancing technologies of these fibres follow those from 1D polymers by improving the orientation along the fibre axis, leading to non-optimized microstructures and low integrated performances. Here, we show a concept of bidirectionally promoting the assembly order, making graphene fibres achieve synergistically improved mechanical and thermal properties. Concentric arrangement of graphene oxide sheets in the cross-section and alignment along fibre axis are realized by multiple shear-flow fields, which bidirectionally promotes the sheet-order of graphene sheets in solid fibres, generates densified and crystalline graphitic structures, and produces graphene fibres with ultrahigh modulus (901 GPa) and thermal conductivity (1660 W m$^{-1}$ K$^{-1}$). We believe that the concept would enhance both scientific and technological cognition of the assembly process of 2D nanosheets.

Graphene, as one of carbon's allotropes, has the highest mechanical properties of Young's modulus (-1100 GPa) and fracture strength (130 GPa); besides, it has the highest thermal conductivity (-5000 W m$^{-1}$ K$^{-1}$) ever reported at room temperature and excellent carrier mobility (200,000 cm$^2$ V$^{-1}$ s$^{-1}$)[1-3]. Macroscopic graphene fibres are supposed to realize the remarkable mechanical and transport properties of single-layer graphene on the macroscale level, benefiting the lightweight-economic target of engineering materials[4,5]. In principle, orderly and densely assembling graphene sheets to giant sp$^2$-hybrized graphitic domain with high crystallinity should achieve both high transport and mechanical properties[4,6-9]. To date, graphene fibres display higher thermal conductivity but inferior Young's modulus

(<400 GPa) and moreover these properties are far below that expected for a single-layer graphene, leaving a challenge to realize high mechanical and thermal properties in graphene fibres.

Previous works have highlighted the importance of improving the axial alignment of graphene sheets to enhance the mechanical and transport properties of graphene fibres, analogy to aligning linear chains in 1D polymer fibres. For examples, shear-flow-induced alignment was developed to fabricate highly aligned graphene assemblies[10-16]. Besides aligning graphene sheets during solution processing, post plastic-stretching was further used to align graphene sheets by eliminating wrinkles and promote overall properties of graphene fibres efficiently[17-21]. However, the 2D topology of single-

[1]MOE Key Laboratory of Macromolecular Synthesis and Functionalization, International Research Center for X Polymers, Department of Polymer Science and Engineering, Zhejiang University, 38 Zheda Road, Hangzhou 310027, P. R. China. [2]Applied Mechanics Laboratory, Department of Engineering Mechanics and Center for Nano and Micro Mechanics, Tsinghua University, Beijing 100084, P. R. China. [3]Key Laboratory for Thermal Science and Power Engineering of Ministry of Education, Department of Engineering Mechanics, Tsinghua University, Beijing 100084, P. R. China. [4]Shanxi-Zheda Institute of Advanced Materials and Chemical Engineering, Taiyuan 030032, P. R. China. [5]These authors contributed equally: Peng Li, Ziqiu Wang, Yuxiang Qi. ✉e-mail: yingjunliu@zju.edu.cn; zhenxu@zju.edu.cn; xuzp@tsinghua.edu.cn; chaogao@zju.edu.cn

layer graphene mismatches with the 1D fibre geometry, which has been largely neglected: the loose assembly of graphene sheets in transverse direction of graphene fibres still exists; furthermore, the increased alignment in axial direction inversely leads to wrinkles and ridges in transverse direction as evidenced in previous reported graphene papers[19]. Such disordered and loose arrangement of graphene sheets in the cross-section heavily reduces the fibre density and crystallinity. This indicates that the aligning strategy for property-enhancement in experience for linear polymer fibres is not favouring to copy for fibres assembled from 2D molecules.

Here, we report a multiple shear-flow assisted wet-spinning (MSW) to bidirectionally promote the assembly order of graphene fibres (Fig. 1), which brings significant improvement in both mechanical properties and thermal conductivity. Our MSW strategy shows the flexibility of tailoring the microstructures by controlling the distribution of shear-flow fields and allows fine manipulation of the ordered assembly of graphene sheets in both longitudinal and transverse directions. Among the tailored microstructures, the transversely concentric and axially aligned sheet-order allows improved crystallization during high-temperature graphitization, affording densified and highly crystalline graphitic structures and improvements in both mechanical property and thermal conductivity. Especially, the prepared graphene fibre even achieves the ultrahigh Young's modulus of a single-layer

graphene. The correlation of MSW assembly process, microstructural and crystalline control, and properties was established and would enhance the scientific comprehension of the assembly process of 2D molecules.

## Results

### Concepts of bidirectionally improved assembly order

Graphene fibre is a one-dimensional macroscopic ordered assembly formed by graphene oxide (GO) liquid crystals under unidirectional shear-flow, following by chemical and thermal reduction[4,11,14,16]. 2D GO sheets flowing in a tubular channel become curved and distorted as a result of the mismatching of the geometries of the sheets and the channel, leading to a loose assembly (Fig. 1e, Supplementary Fig. 1a). Bidirectionally promoting assembly order refers to simultaneously optimizing sheet-order in fibre cross-section and sheet alignment along fibre axis by our MSW technology.

Typically, three stages of sheet-order in the MSW process are shown in Fig. 1c. Pre-ordered GO sheets in anisotropic liquid crystals are directed under tubular shear-flow to form aligned sheet-arrangement along the flow direction as schemed as Plane I (Fig. 1d), yet disordered sheet-order in transverse cross-section retains and aggregates under unidirectional tubular flow velocity $u$ (Fig. 1e). Plane II demonstrates that the introduced rotating shear-flow compels

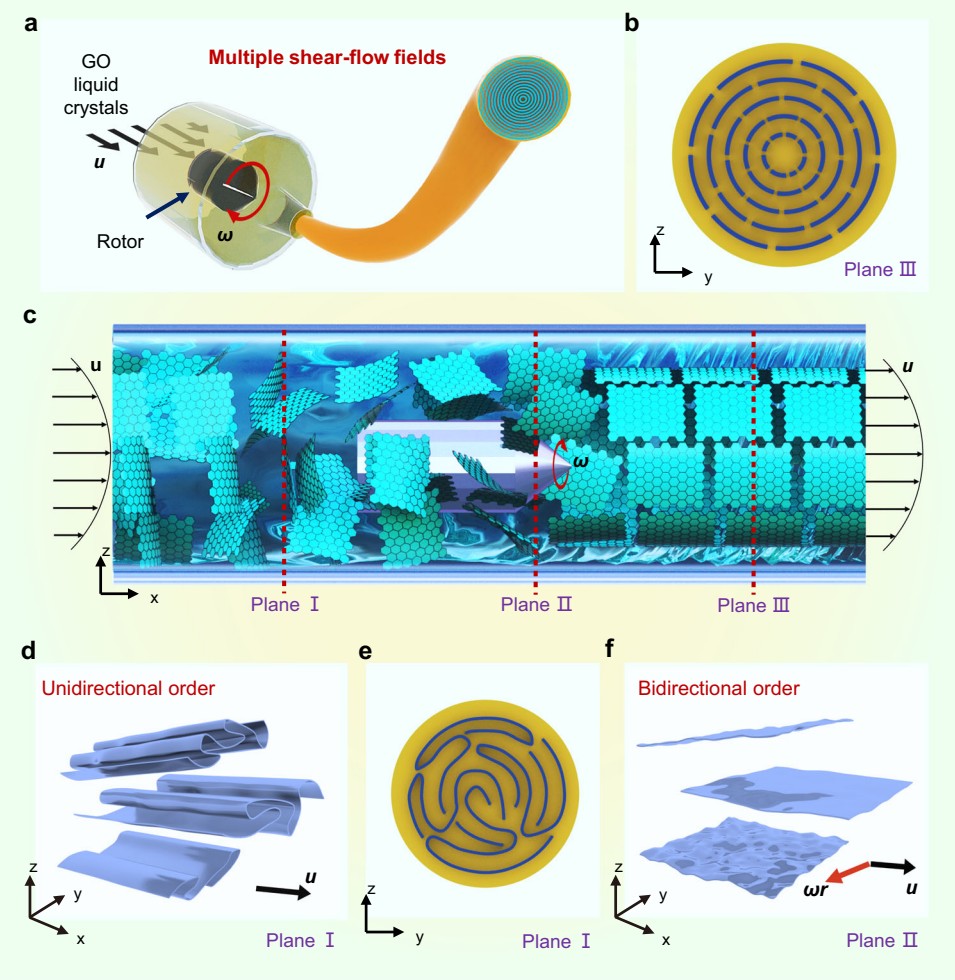

**Fig. 1 | Schematic illustration of the concept of bidirectionally promoting graphene assembly order. a** GO fibres fabricated by multiple shear-flow fields combining tubular and rotating shear. **b**–**f** Structural illustration of the sheet-order in the whole spinning tube under the multiple flow fields (**c**), in which Plane I (**d**, **e**) depicts the aligned sheets along the flow direction but wrinkled conformation perpendicular to the flow direction under unidirectionally tubular flow field, Plane II (**f**) shows the aligned sheets in both the flow direction and transverse cross-section under the multiple flow fields, Plane III (**b**) illustrates the optimally concentric structure after bidirectionally promoting assembly order.

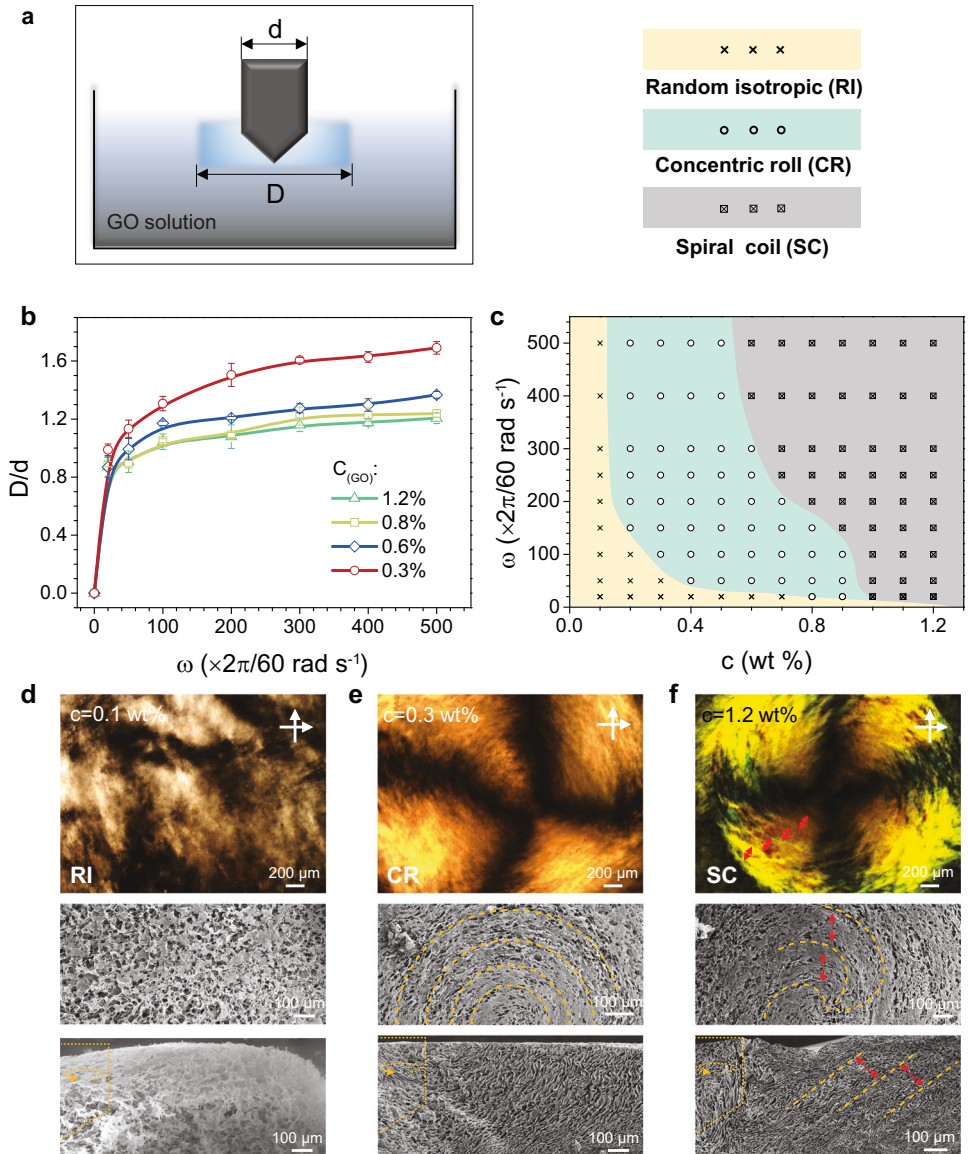

**Fig. 2 | Static observation of GO liquid crystals under rotating shear field.**
**a** Schematic illustration of the static observation. *d* refers to the diameter of the rotating rotor, while *D* refers to the diameter of the area affected by the rotating shear field. **b** Relationship of *D/d* and rotating angular velocity at different GO concentration *c*. Error bars represent s.d. of the measured *D/d*. **c** Phase diagram of GO liquid crystal texture at variable GO concentration and rotating angular velocity $\omega$. **d–f** The polarizing optical microscope (POM) images and surface and cross scanning electron microscope (SEM) images of the three typical phases, including random isotropic phase (**d**), concentric roll (**e**), and spiral coil (**f**).

graphene sheets to extend state under a constant angular velocity $\omega$ (Fig. 1f). Linear velocity $v$ can be described by $v = \omega r$, where $r$ is the distance from the rotor centre to graphene sheets. $v$ attenuates at the position away from central rotor, which forms a rotating shear-flow field[22]. Dense and ordered structure after bidirectionally promoting assembly order is shown in Plane III (Fig. 1b).

## Static observation of GO liquid crystals under rotating shear-flow field

The transformation of GO liquid crystalline textures was firstly observed under the single rotating shear-flow, in which the rotating angular velocity $\omega$ of the introduced rotor and concentration $c$ of GO liquid crystals are crucial for the formed texture. The area affected by rotating field was determined (Fig. 2a, b; *D* is defined as the diameter of the affected round area; *d* is the diameter of the rotating rotor, $d = 2r$). Increased rotating angular velocity $\omega$ and decreased GO concentration result in enhanced *D/d*, which directs the choice of rotor diameter.

Liquid crystals of GO solution were determined by the concentration of GO solution[23]. GO solution shows isotropic phase at low GO concentration. In this concentration range, GO sheets with low molecular rotation energy barrier distribute randomly and move separately, and any ordered texture by rotating shear-flow is unstable (Fig. 2d). As concentration increases, the translational entropy increase compensates the orientation entropy loss, leads to the formation of thermal dynamic stable liquid crystalline mesogens[24,25]. In higher concentration range, strong excluded volume repulsion retards the relaxation of liquid crystalline mesogens and keeps the concentric liquid crystalline texture reserved (Fig. 2e, Supplementary Fig. 2)[25]. When GO concentration exceeds 1.0 wt%, steric hindrance from the overlap of excluded volume repulsion endows the mesogens with powerful elastic energy for resistance to deformation and spiral liquid crystals with periodic bands form (Fig. 2f)[14,26]. Rotating angular velocity $\omega$ also influences the sheet-arrangement. We concluded a phase diagram in Fig. 2c.

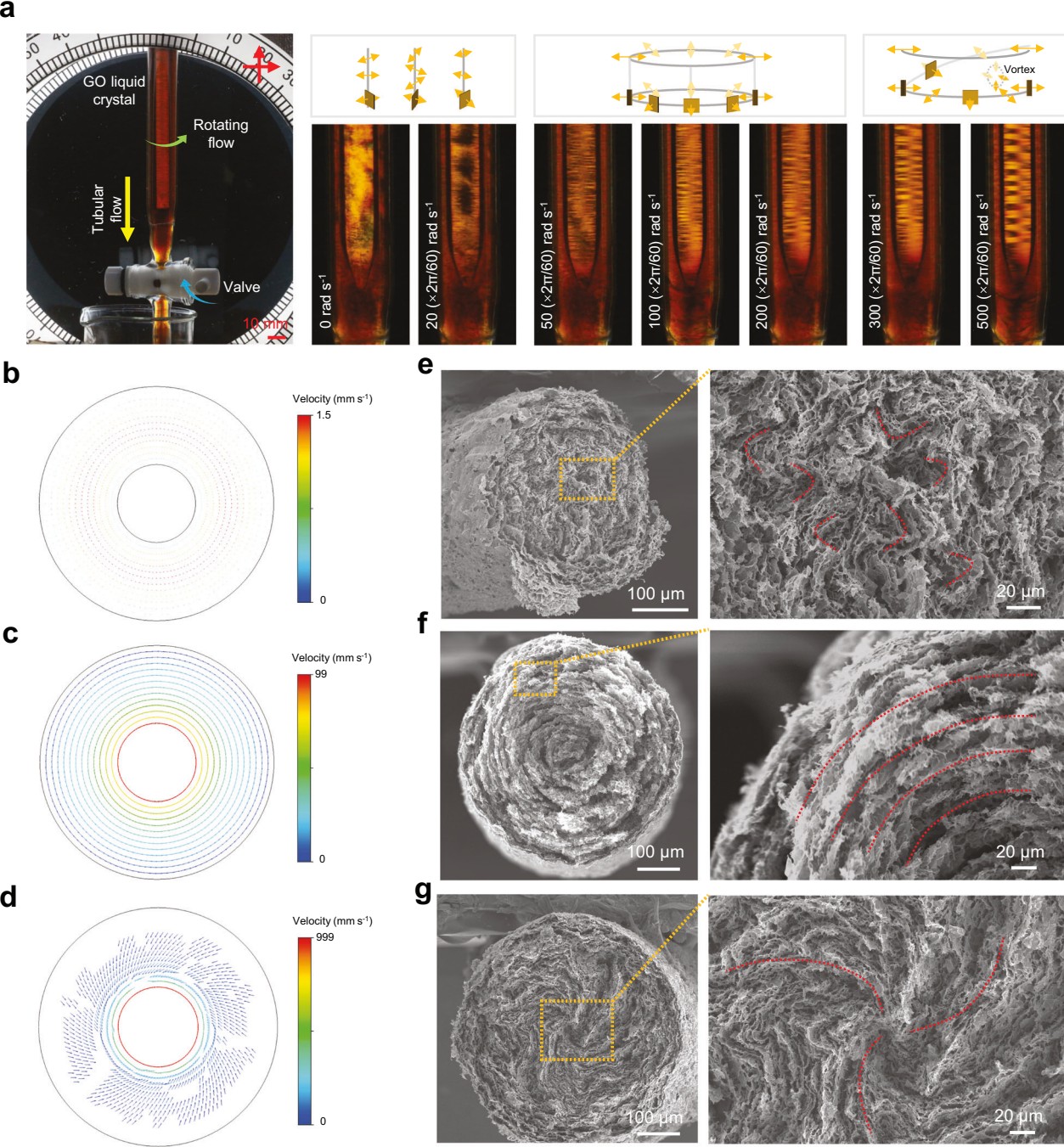

**Fig. 3 | Formation of the three sheet-orders. a** Dynamic observation of GO liquid crystalline textures and schematics of sheet-orders under multiple shear-flow fields. Secondary vortex forms in the spiral textures, resulting in periodic disorders. **b** Velocity distribution on the cross-section of unidirectional tubular shear-flow field. **c, d** Velocity distribution on the cross-section of multiple shear-flow fields at moderate (**c**) and overhigh (**d**) rotating angular velocities, illustrating the formation mechanism of the variable sheet-orders. **e**–**g** SEM images of random structure of aerogel fibre cross-section without rotating shear (**e**), concentric textured cross-section at rotating angular velocity $\omega$ of 100 (×2π/60) rad s$^{-1}$ (**f**), spiral textured cross-section at rotating angular velocity $\omega$ of 500 (×2π/60) rad s$^{-1}$ (**g**).

## Liquid crystalline textures under multiple shear-flow fields

The transformation process of assembly order of graphene sheets at the MSW fields depicts tuneable sheet-order in the cross-sections (Supplementary Fig. 3). We tracked the GO liquid crystals and analysed the shear force and velocity distribution under multiple shear-flow fields (Fig. 3a–d and Supplementary Fig. 4). GO liquid crystals flow along the tube axis under unidirectional tubular shear force without radial ordered texture (Fig. 3a, b, Supplementary Fig. 4a₁, a₂, a₃, and Supplementary Movies 1, 2). But if rotating flow is introduced, regular textures intuitively appear. When a moderate angular velocity $\omega$ is applied, a stable concentric velocity field forms under the influence of both rotating and tubular shear forces, yielding a concentric liquid crystalline texture (Fig. 3a, c, Supplementary Fig. 4b₁, b₂, b₃, and Supplementary Movie 3). When the angular velocity $\omega$ is further improved, excessive centrifugal forces make the radial pressure gradient and the viscous forces unable to dampen out disturbances in the flow, causing a secondary vortex velocity field (Fig. 3a, d, Supplementary Fig. 4c₁, c₂, c₃). In this condition, spiral GO liquid crystalline texture with periodic disorders is clearly observed (Fig. 3a and Supplementary Movie 4), corresponding to the spiral sheet-arrangement

in Fig. 2f. The gradient rotor used in multiple shear-flow fields protects the concentric structure in the core region from destroying. As shown in the velocity distribution in Supplementary Fig. 5a, c–g, the concentric structure forms following three steps illustrated by the four sections. GO sheets are initially arranged in concentric texture under multiple shear-flow fields with rotating rotor in central region (Section 1 in Supplementary Fig. 5d). Then with the reduced diameter of rotating rotor, GO sheets around the rotor converge at the tip, in which GO sheets are suffering continued rotating shear forces to keep the concentric texture in the core region (Sections 2 and 3 in Supplementary Fig. 5e, f). GO liquid crystals finally flow along the axis in the spinning tube (Section 4 in Supplementary Fig. 5g). Converged GO sheets in liquid state at both outer region and core region are aligned along the tube axis based on the position-resolution small-angle X-ray scattering (SAXS) results (Supplementary Fig. 5h–j).

GO aerogel fibre was systematically fabricated by our MSW technology and characterized using freeze-drying and scanning electron microscopy (SEM) techniques[27]. As shown in Supplementary Figs. 6 and 7, we fabricated variable GO aerogel fibres by tuning GO concentrations $c$ and rotating angular velocity $\omega$. The sheet-orders in these fibre cross-sections typically involve randomly distributed GO sheets, GO sheets arranged with a concentric texture and a spiral texture, which are consistent with the liquid crystalline tracking of random isotropic, concentric roll, and spiral coil in Figs. 2 and 3a. Among the graphene structures prepared with GO concentration of 0.6 wt%, GO aerogel fibres shows random sheet-order at $\omega$ lower than 20 ($\times 2\pi/60$) rad s$^{-1}$; concentric roll roughly at $\omega$ of 50, 100, and 200 ($\times 2\pi/60$) rad s$^{-1}$; spiral coil at $\omega$ higher than 300 ($\times 2\pi/60$) rad s$^{-1}$ (Supplementary Fig. 6). We thus chose GO concentration of 0.6 wt% for deeply investigating the fibre performance at different angular velocity $\omega$ covering the three typical sheet-orders (Fig. 3e–g).

The thermal conductivity of the graphene aerogel fibres with random, concentric, and spiral sheet-orders after annealing at 2700 °C were tested. Graphene aerogel fibres with concentric sheet-order show the optimized assembly order and have the highest electrical and thermal conductivities up to 2504.7 S m$^{-1}$ and 19.7 W m$^{-1}$ K$^{-1}$ (Supplementary Fig. 8a and Supplementary Movie 5). These graphene aerogel fibres structured with different sheet-orders could provide platforms for diverse functional applications, such as energy storage and conversion. We exhibited an application of a phase-change graphene material by filling the concentric graphene aerogel fibre with polyethylene glycol (PEG, 94 wt%). The high thermal/electrical conductivities enable the phase-change functional fibre to heat up to 80 °C as quick as 3 s at external electron stimuli of only 1.5 V (Supplementary Fig. 8b, c), superior to the performance of previous reports[28].

## Fabrication of continuous graphene fibres by MSW technology

Decreasing the microcapillary size from 1500 to 150 μm has little effect on the manipulated sheet-arrangement as shown in Supplementary Fig. 9. We also found that the introduced rotating shear-flow has little influence to the stability of MSW process for continuously fabricating GO fibres, even at angular velocity $\omega$ as high as 500 ($\times 2\pi/60$) rad s$^{-1}$ (Supplementary Fig. 10b). Thus, continuous graphene fibres by the MSW technology at different $\omega$ covering the three sheet-orders were fabricated following by chemical and thermal reduction (Supplementary Fig. 11f). For clarity, fibres derived from the three sheet-orders are denoted as random, concentric, and spiral fibres, respectively.

## Crystalline analysis of graphene fibres

Highly densified and crystalline graphitic structures were formed in concentric graphene fibres by bidirectionally promoting the assembly order (Fig. 4). Concentric sample shows the highest value of density reaching 2.02 g cm$^{-3}$, in accordance with the smooth and dense stacking in the fibre cross-section from concentric sheet-order (Fig. 4f, Supplementary Figs. 1b and 12, and Supplementary Table 2). As a comparison, graphene fibres originated from the random sheet-order have numerous acute loose stacking and lower density of 1.89 g cm$^{-3}$ (Supplementary Fig. 1a). Atomic defects on GO sheets are restored during thermal annealing (Supplementary Fig. 13)[11,16,17]. Axial and transverse slices of graphene fibres were observed using transmission electron microscopy (TEM)[29–32]. As shown in Fig. 4a–c, huge perfect graphitic crystallites grow in both axial and transverse directions after the thermal graphitization at 2700 °C, where three-dimensional crystalline sizes and orientation order ($f$) determine the crystallinity, including the thickness ($Lc$), longitudinal length ($La_{\parallel}$), and transverse length ($La_{\perp}$) (Supplementary Fig. 14).

The crystalline factors were quantitatively evaluated by wide-angle X-ray scattering (WAXS) analysis. 2D WAXS patterns of graphene fibres show two symmetrical (002) scattering rings in the equatorial direction and annular (100)/(101) broad peaks (Fig. 4d)[17]. The integrated curves of all samples at different $\omega$ show these three scattering peaks, where (002), (100), and (101) peaks position at 21.3°, 34.1°, and 35.6° (Wavelength of X-ray is 0.124 nm, Supplementary Fig. 15). The corresponding azimuthal angle $\varphi$ plot of (002) plane features a sharp peak at $\varphi = 0°$, indicating the orientation order parameter $f$ (Fig. 4d–f, Supplementary Fig. 16b)[17,33,34]. Three-dimensional crystallite sizes $Lc$, $La_{\parallel}$, and $La_{\perp}$ were calculated from the (002) and (100) peaks in equatorial and meridional scanning curves, respectively (Fig. 4g, Supplementary Fig. 16c, d, see Characterization)[35]. We showed that optimized concentric texture at $\omega = 100$ ($\times 2\pi/60$) rad s$^{-1}$ is in favour to form ordered assembly in both longitudinal and transverse directions in the final dense concentric graphene fibre, which results in an improved order parameter (0.93) and enlarged crystallite sizes ($Lc = 68.5$ nm, $La_{\parallel} = 236.6$ nm, $La_{\perp} = 114.1$ nm). Especially, the introduced rotating shear-flow improves the sheet-order in the fibre cross-section compared to previously reported graphene fibres, thus distinctly facilitating the increasement of thickness ($Lc$) and transverse length ($La_{\perp}$) of graphitic crystallites (Fig. 4g, Supplementary Fig. 16d). Concentric graphene fibre has increase rates of $Lc$, $La_{\perp}$, and $La_{\parallel}$ reaching 235%, 74%, and 31% compared to those of random samples without MSW technology, respectively. Apparent crystalline sizes were also collected to verify the enhanced crystallinity from TEM images (Supplementary Fig. 17)[36]. The statistically apparent length of the optimized graphene fibre reaches 294.8 nm in TEM images. Higher or lower $\omega$ leads to the formation of either spiral disorders or random disorders, respectively, causing inferior orientation order and smaller crystallite sizes.

## High-performance graphene fibre

Optimized sheet-order of graphene fibre achieves the densified and crystalline graphitic structure, thus improving the mechanical and functional properties. Tensile tests showed that the Young's modulus of the concentric graphene fibre had a huge increase reaching 642 GPa (highest value 833 GPa), higher than that of the random graphene fibre and spiral graphene fibre (Fig. 4h, Supplementary Fig. 18b, Supplementary Table 2)[10,11,16,17]. The load transfer efficiencies of concentric fibres and random fibres were evaluated by an in situ Raman test, in which the downshift of G-band denotes the deformation of graphene. The G-band shift rate of concentric graphene fibres is 17.5 cm$^{-1}$ per 1% strain, 113% higher than that of random fibres, indicating an effective load transfer in concentric fibres (Supplementary Fig. 19). The improved density and optimized crystalline order jointly contributed to the record-high value of Young's modulus (Fig. 4i, Supplementary Table 2)[16,17].

The average tensile strength of concentric graphene fibres is 2.9 GPa, lower than that of random fibres, which results from the increased defect-sensitivity of concentric fibres since cracks propagate along the enlarged $Lc$ and $La_{\perp}$[37,38]. Defects inside graphene fibres mainly involve misoriented grain boundary and microvoids (Supplementary Fig. 20). Misoriented grain boundary can be reduced by the

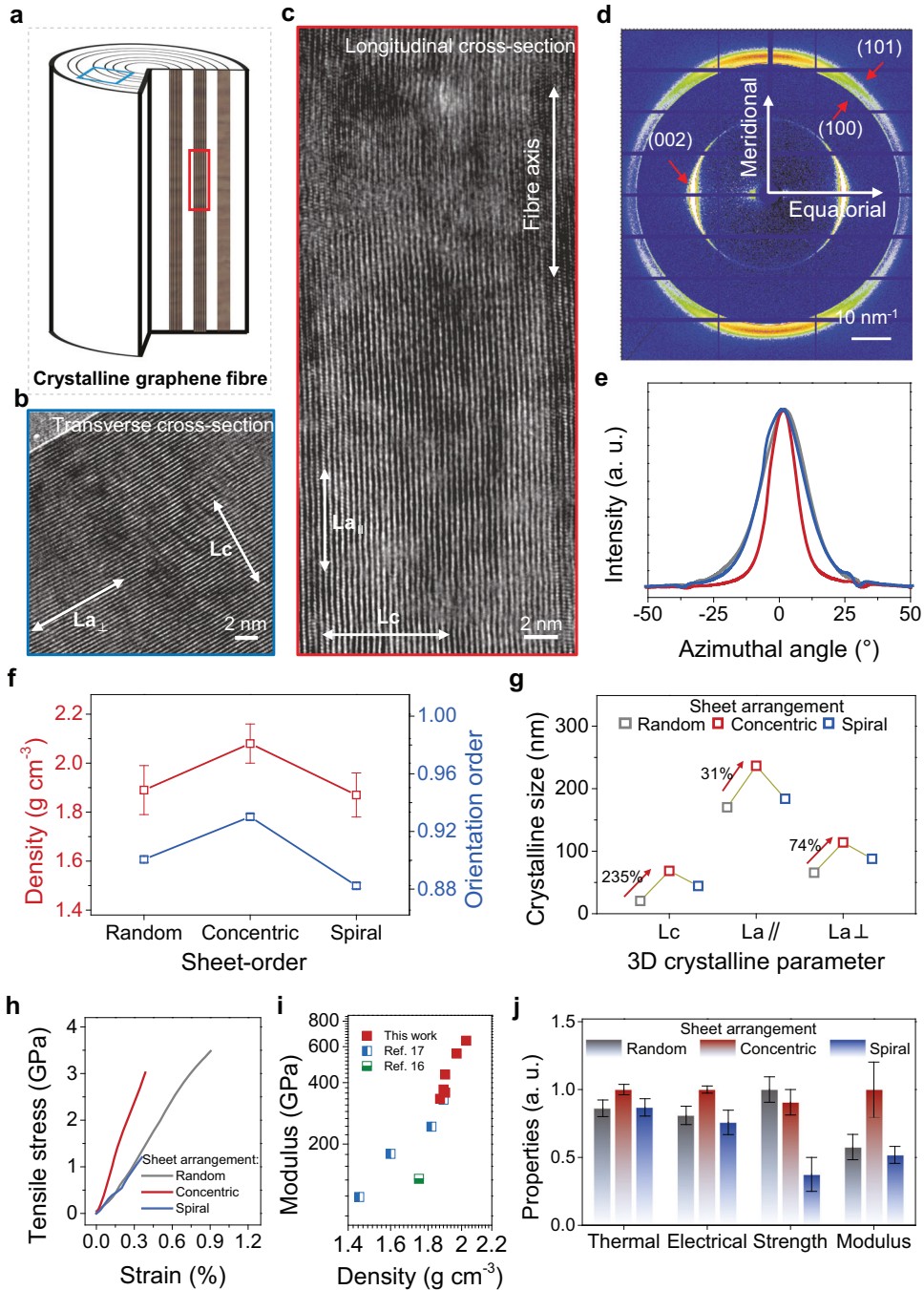

**Fig. 4 | Crystalline characterization of the prepared graphene fibres.**
**a** Schematic depicting the ideal crystalline graphene fibre. Note, the solid fibres deviate from a concentric texture on the cross-section due to the severe shrinkage during solidification. **b, c** Transmission electron microscope (TEM) images of a graphene fibre along transverse (**b**) and longitudinal (**c**) direction, illustrating the three-dimensional crystalline structure in a single fibre. **d** Wide-angle X-ray scattering pattern of the concentric graphene fibre, showing the high crystallinity. **e** The azimuthal scanning curves of the random (grey line), concentric (red line),

and spiral (blue) graphene fibres. **f** Density and orientation order of the prepared graphene fibres. **g** The crystalline size of the 3D graphitic crystallites. **h** The typical tensile stress-strain curves of the prepared graphene fibres. **i** Relationship between the Young's modulus and density of graphene fibres. **j** Overall properties of the fabricated graphene fibres, including thermal and electrical conductivities, tensile strength, and Young's modulus. Error bars in (**f**), (**j**) represent s.d. of the measured properties.

improved orientation. Microvoids form mainly in the huge shrinkage during solidification and randomly position at the gap of folds without distinction in outer and inner regions of fibre structure as shown in TEM images and 3D constructed image of fibre transverse cross-section (Supplementary Figs. 21 and 22). We quantitatively measured the microvoid defects by SAXS[39]. As shown in Supplementary Fig. 23, the content of microvoids in concentric graphene fibres decreases,

which is in agreement with the enhanced density. But the microvoid defect size of the concentric graphene fibre is enlarged, which may also lead to the decreased tensile strength[40].

Besides the increased Young's modulus, dramatically enlarged three-dimensional crystallite sizes also enhance the transport of the phonons and electrons (Fig. 4j). The thermal conductivity of the concentric graphene fibre reaches 1590 W m$^{-1}$ K$^{-1}$, about 16% higher than

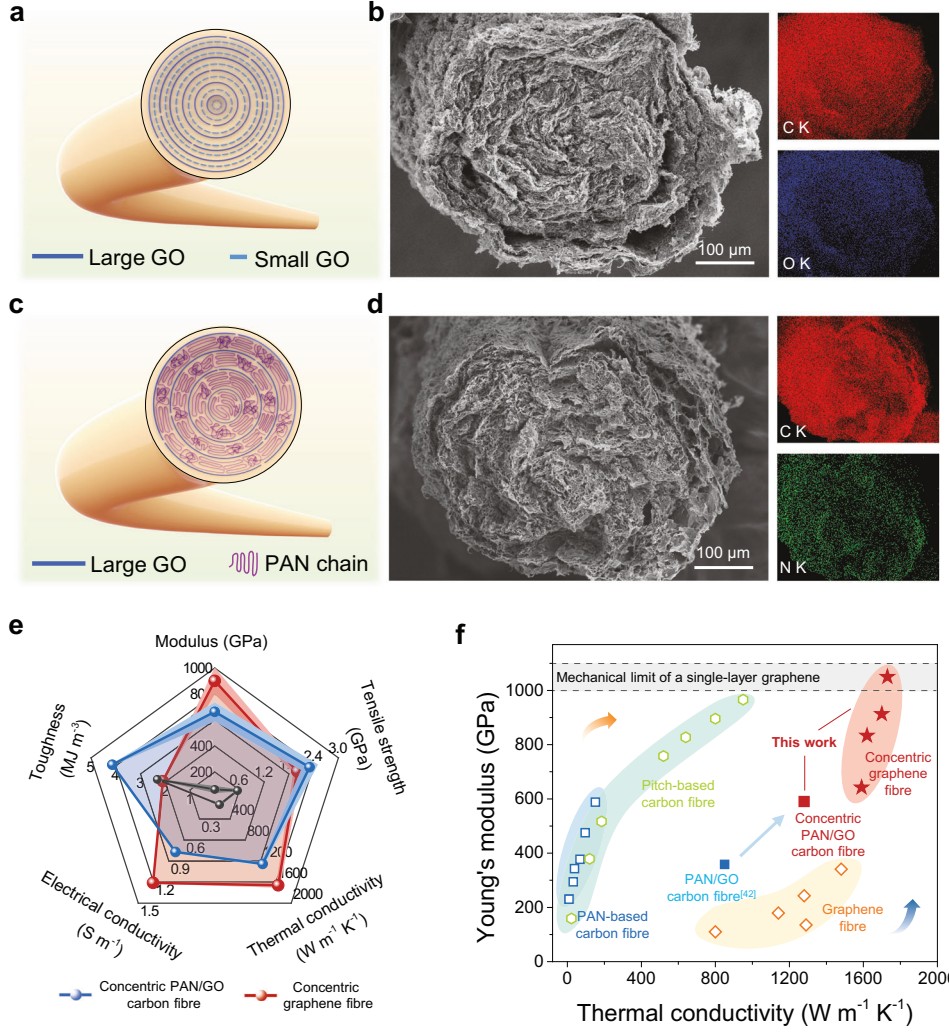

**Fig. 5 | Concentric skeleton of graphene sheets for high-performance carbonaceous fibres. a** Schematic depicting the small-sized GO is filled in the concentric skeleton of large-sized GO. **b** SEM image and energy dispersive spectroscopy (EDS) patterns of the aerogel fibre with 70 wt% large-sized GO and 30 wt% small-sized GO. **c** Schematic depicting the PAN chains are intercalated in the interlayer of concentric GO sheets. **d** SEM image and EDS patterns of the aerogel fibre with 70 wt% PAN chains and 30 wt% GO sheets. **e** Mechano-thermo properties of the concentric graphene fibre and PAN/GO carbon fibre. The black points are the properties of the fabricated pure PAN carbon fibre. Error bars represent s.d. of the measured properties. **f** Ashby plots of the Young's modulus and thermal conductivity, including the fabricated concentric graphene fibres, previously reported graphene fibres, and PAN-based/Pitch-based carbon fibres.

that of the random graphene fibre and spiral graphene fibre. (Supplementary Fig. 18c). The electrical conductivity of the concentric graphene fibre was optimized as high as $1.2 \times 10^6$ S m⁻¹ (Supplementary Fig. 18d).

**Generality of the ideal concentric skeleton model**
We further demonstrated that the concentric structure of graphene sheets was an ideal skeleton model for fabricating carbonaceous fibres with high modulus and excellent thermal conductivity (Fig. 5a, c). Small-sized GO sheets were reported to fill the microvoids in graphene fibres assembled by large-sized GO sheets, affording densified graphene fibres with improved thermal conductivity and Young's modulus[16]. Concentric graphene fibres composing of the same content of small-sized GO sheets (30 wt%) were fabricated by the MSW strategy. As shown in Fig. 5b, by fixing the total mass fraction of large-sized and small-sized GO sheets to 0.6 wt% and rotating angular velocity of 100 (×2π/60) rad s⁻¹, GO aerogel fibre shows distinctly concentric texture. The prepared concentric graphene fibre exhibits a high Young's modulus of 901 GPa (highest value 1030 GPa), about 6 times higher than that of the reported fibre (Fig. 5e, Supplementary Fig. 24a,

and Supplementary Table 3)[16]. The highest value even reaches the mechanical limit of single-layer graphene (Fig. 5f)[3]. The thermal and electrical conductivity were measured to be 1660 W m⁻¹ K⁻¹ and $1.21 \times 10^6$ S m⁻¹, respectively (Supplementary Fig. 24a). The thermal conductivity of concentric graphene fibres achieves near 75% of a benchmark of highly oriented pyrolytic graphite[17,41]. Such graphene fibres with densified and crystalline graphitic structures achieve the integrated mechano-thermo properties, superior to those of most commercial pitch-based carbon fibres that are well-known for their highly thermal conductivity and Young's modulus (Fig. 5f). Giving the lower density of graphene fibres, the specific thermal conductivity and specific modulus are ~0.8 W m⁻¹ K⁻¹/(kg m⁻³) and ~0.4 GPa/(kg m⁻³), outperforming most conventional metal materials (Supplementary Fig. 25 and Supplementary Table 4). Such high thermal property, combining with high modulus, makes this fibre an important candidate engineering material in high-performance composites.

The concentric structure of graphene sheets also directed the preparation of highly thermal conductive and stiff polyacrylonitrile (PAN)-based carbon fibres. For commercial PAN-based carbon fibres, inferior graphitic structures make the fibres exhibit poor thermal

conductivity (~32 W m$^{-1}$ K$^{-1}$). We showed that the concentric roll texture of GO liquid crystals still formed even with the addition of 70 wt% PAN chains (Supplementary Fig. 26). PAN/GO carbon fibres with 70 wt% PAN and 30 wt% GO were then prepared by the MSW strategy. PAN chains were uniformly distributed in the concentric skeleton of GO sheets (Fig. 5d). Improved crystallinity of PAN-based carbon fibres can be realized by confining graphitization with the addition of GO sheets[42]. The concentric PAN/GO carbon fibres show improved thermal conductivity and Young's modulus reaching 1254 W m$^{-1}$ K$^{-1}$ and 663 GPa, about 47% and 85% higher than those of previously reported fibres without optimized sheet-order (Fig. 5e, f, Supplementary Fig. 24b, and Supplementary Table 3). The thermal conductivity of the prepared concentric PAN/GO carbon fibre is ~38 times higher than that of commercial PAN-based carbon fibre and even exceeding the commercial pitch-based carbon fibre, making it a competitive material applicable for low-cost thermal-managing conditions (Fig. 5f). Based on these results, concentric sheet-order of 2D molecules would be an ideal structure model for fibre materials with highly integrated mechano-thermo properties.

## Discussion

Bidirectionally promoting the sheet-order of graphene fibre has been realized to achieve both highly thermal conductivity and excellent modulus. The established MSW technology was demonstrated to fine control the assembly order of graphene sheets in both longitudinal and transverse directions, thus offering a unique opportunity to fabricate macroscopic graphene structures with variable sheet-arrangement, such as random, concentric, and spiral sheet-orders. These flexible macroscopic graphene structures could be novel platforms for diverse functional applications. The transversely concentric and axially aligned sheet-order optimizes the densified and crystalline graphitic structures, being an ideal structure model of graphene sheets for achieving fibres with highly thermal conductivity and Young's modulus. This concept of bidirectionally promoting sheet-order may be extended to other nanoparticles with planar anisotropic structure for producing high-performance macroscopic materials.

## Methods

### Preparation of GO and PAN/GO spinning dope

To obtain a GO/N, N-dimethyl formamide (DMF) solution for wet-spinning, the water in purchased GO aqueous (average lateral size of 105.2 μm and 5.1 μm as shown in Supplementary Fig. 10a–e, Hangzhou Gaoxi Technology Co. Ltd) solution was replaced by DMF (Sinopharm Chemical Reagent Co. Ltd) via a repeating centrifugation method for at least five times to obtain concentrated GO/DMF liquid crystal solution. Before wet-spinning for fabricating graphene fibres, GO/DMF dope should be treated by removing possible impurities and degassing adequately.

To prepare the GO spinning dope with 70 wt% large-sized GO and 30 wt% small-sized GO, large-sized and small-sized GO solutions with concentration of 0.6 wt% were mixing with a mass ratio of 7:3. The GO mass fraction in the spinning dope is 0.6 wt%.

To prepare the PAN/GO spinning dope with 70 wt% PAN and 30 wt% GO, PAN (molecular weight 250000, Sigma-Aldrich) powder were dissolved in DMF with a mass fraction of 2.8 wt%. The concentration of GO DMF solution was tuned to be 1.2 wt%. Mixing equal parts of PAN and GO solutions obtains the PAN/GO spinning dope with total GO mass fraction of 0.6 wt%.

### Fabrication of the macroscopic graphene structure with variable textured cross-section

Macroscopic graphene structures with variable sheet-orders were prepared by our MSW method and freeze-drying technology. Typically, GO aqueous solution with different concentrations was extruded to a home-made rotary extruder, where GO sheets suffer from both tubular shear for axial aligning and rotating shear for transversely ordered assembling. Then GO aqueous solution was spun to 3 wt% CaCl$_2$ (Sinopharm Chemical Reagent Co. Ltd) aqueous solution to form a stable gel fibre. After washing the obtained gel fibre using deionized water for at least three times, the gel fibre was freeze-dried to fabricate solid state fibre with variable sheet-order[43]. Specially, the diameter of GO gel fibres was controlled by tuning the diameter of spinneret, and aerogel fibres with arbitrary diameter showed corresponding sheet-order in fibre cross-section at the specific rotating angular velocity as shown in Supplementary Fig. 8. PAN/GO gel fibre were fabricated by extruding the PAN/GO DMF solution to 3 wt% CaCl$_2$ aqueous solution with the MSW technology.

Macroscopic graphene aerogel structures were prepared by chemical reducing and thermal annealing the GO aerogel fibres as shown below in the section of fabrication of graphene fibres. The graphene aerogel fibre-based phase-change materials were fabricated by immersing the graphene aerogel fibre into PEG (Sinopharm Chemical Reagent Co. Ltd) melt at 80 °C in a vacuum oven for 3 h to ensure that the aerogel fibre was infused with PEG. Then the sample was allowed to hang under 80 °C to remove the excess PEG adhering on the fibre surface.

### Fabrication of graphene fibre and PAN/GO carbon fibre

Following the established MSW technology, GO/DMF spinning dope (0.6 wt%) was injected through a spinneret (80 μm diameter) into coagulation baths containing a mixture of DMF and ethyl acetate (Sinopharm Chemical Reagent Co. Ltd). GO fibres were solidified and collected continuously onto graphite rollers, and plasticization-stretched continuously using a mixed bath of acetic acid and water to restrain the drying-induced shrinkage.

GO fibres fixed on the graphite rollers under tension were first reduced by hydroiodic acid (HI, Sinopharm Chemical Reagent Co. Ltd) at 90 °C for 12 h. Then the reduced GO fibres fixed on the graphite rollers were annealed using a tube furnace. The samples were heated from room temperature to 1300 °C at a rate of 1 °C min$^{-1}$ and kept at 1300 °C for 1 h in a flow of hydrogen/argon (20 vol%) mixture. Finally, the samples were heated from room temperature to 2700 °C at a rate of 10 °C min$^{-1}$ and maintained at 2700 °C for 1 h in a flow of argon.

To fabricate PAN/GO carbon fibre, PAN/GO spinning dope was injected into the coagulation bath of pure ethyl acetate with the established MSW technology, followed by plasticization-stretching in a plasticization bath (H$_2$O/DMF)[42]. The PAN/GO fibres are collected continuously onto a graphite roller for subsequent thermal treatments.

### Static observation of GO liquid crystals under rotating shear field

Concentrated GO/DMF solution was diluted to a gradient concentration. Then GO/DMF solutions with different concentrations were flat filled in a home-made liquid crystal observation cell. After applying given rotating angular velocity to the GO solution, the cell was fixed under POM to observe the liquid crystal texture. GO liquid crystal solution was also freeze-dried to solid state. The solid foam was characterized under SEM along the surface and cross-section, respectively.

### Dynamic tracking of GO liquid crystals under multiple shear-flow fields

GO/DMF solutions (0.6 wt%) were filled in a home-made liquid crystal observation tube, which possessed a valve for tubular flow and a glass rod with a gradient tip for rotating flow. This observation tube was fixed between two orthogonal polarizers. After applying given rotating angular velocity to the GO solution and opening the valve, specific GO liquid crystalline textures were recorded by a digital camera.

## Characterizations

The static observation of liquid crystal under rotating shear field was conducted by polarizing optical microscope (Nikon, LV100N POL). The morphology and microstructure of macroscopic graphene structures and graphene fibres were characterized by field-emission scanning electron microscope (Hitachi, S4800). We also collected GO size under scanning electron microscope and analysed its distribution. 3D constructed image of solid graphene fibre are conducted on Helios G3 UC. The crystalline microstructures of graphene fibres and graphene stacking in single graphitic crystallite were characterized by transmission electron microscope (Tecnai F20 FEI and Titan G2 60-300 FEI). The slice samples along both axial and transverse directions used for TEM measurement were fabricated by ion-milling[34]. Differential scanning calorimetry (Q100 TA) was carried out to characterize the enthalpy of the graphene aerogel fibre-based phase-change materials. Wide-angle X-ray scattering (WAXS) measurement was used to determine the three-dimensional crystalline sizes[17,35] (thickness of graphitic crystallites $Lc$, longitudinal length $La_{\parallel}$, and transverse length $La_{\perp}$) according to Scherrer equation $L = K\lambda/(\beta \cos \theta)$, where $K = 0.89$ for $Lc$, $K = 1.84$ for $La_{\parallel}$ and $La_{\perp}$, $\lambda$ is the source wavelength of 0.124 nm, $\beta$ (rad) is full width at half maximum of (002) along equatorial direction for $Lc$, (100) along meridian direction for $La_{\parallel}$, and (100) along equatorial direction for $La_{\perp}$. Orientation order parameter was also quantified by azimuthal angle scanning by the (002) reflection in WAXS patterns. SAXS was used to analyse the microvoids in graphene fibre (Supplementary Text 1.3). Position-resolution SAXS (spot diameter 200 μm) was carried on to characterize the arrangement of GO sheets in liquid state at both outer and core regions of ultrathin quartz tube (diameter 1000 μm). WAXS and SAXS tests were carried out on BL16B1 beam line station in Shanghai Synchrotron Radiation Facility. All data were collected by deducting the background scattering from air. For WAXS measurements, the fibres were measured as aligned multifilament bundles for the transmission test mode. Microcomputed tomography (Carl Zeiss, 520 Versa) was used to confirm the concentric texture of graphene fibre. The density of each sample was tested by the sink-float method following previous literatures[16,32], in which three types of liquids, tetrabromoethane ($Br_2CHCHBr_2$), carbon tetrachloride ($CCl_4$) and hexamethylene were used. In situ Raman testing was operated by Renishaw in Via-Reflex Raman microscopy (excitation wavelength of 532 nm) accompanied with a translation stage (PI, M-112.2DG1) whose accuracy is 250 nm. Tensile stress-strain tests were conducted using a Keysight T150 UTM with 5 mm gauge length and $1.67 \times 10^{-2} s^{-1}$ extension rate. At least five samples of each type of fibre were tested to measure the average mechanical properties and standard deviations. The electrical conductivity was measured by a standard four-probe method using Keithley 2611B. The thermal conductivity of graphene fibres was measured by a steady self-heating method (Supplementary Fig. 27)[16,44] and examined by a well-established T-type method[45]. Three samples fabricated at variable angular velocities were tested to calculate the average electrical and thermal properties and corresponding standard deviations. The thermal conductivity of graphene aerogel fibres was also tested by the steady self-heating method.

## Data availability

The data that support the findings of this study are available from the corresponding author upon reasonable request.

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

## Acknowledgements

We thank the members of staffs at Shanghai Synchrotron Radiation Facility for assistance in WAXS and SAXS characterizations and Jian-sheng Guo in the Centre of Cryo-Electron Microscopy (CCEM), Zhejiang University for his technical assistance on 3D construction of graphene fibre by SEM and focused-ion-beam (FIB). This work is supported by the National Natural Science Foundation of China (Nos. 52090030, 52090031, 52090032, 51973191, 52122301, 52272046), the Natural Science Foundation of Zhejiang Province (LR23E020003), the Funda-mental Research Funds for the Central Universities (Nos. 226-2023-00023, 2021FZZX001-17), Postdoctoral Research Program of Zhejiang Province (ZJ2022079), National Key Research and Development Pro-gram of China (2022YFA1205300), the Natural Science Foundation of Zhejiang Province (LR23E020003), "Pioneer" and "Leading Goose" R&D Program of Zhejiang 2023C01190, the Research Fund for Nanjing University of Aeronautics and Astronautics (INMD-2021M06), Shanxi-Zheda Institute of New Materials and Chemical Engineering (2021SZ-FR004, 2022SZ-TD011, 2022SZ-TD012), and the International Research Centre for X polymers.

## Author contributions

P.L., Y.L., Z.X. and C.G. designed the research. P.L. and Z.W. carried out the experiment and characterizations. Y.Z. and Z.P.X. did the simulation of graphene distribution in multiple flow fields and the fibre formation. Z.L. and W.M. measured the thermal conductivity of graphene fibres using T-type method. P.L., Z.W. and Y.Q. prepared the macroscopic graphene structure and graphene fibre-based phase-change materials and measured the thermal/electrical conductivities. X.M. and K.S. pro-vided help on mechanical and thermal tests. P.L., G.C. and H.L. fabri-cated the carbonaceous fibres based on the concentric graphene skeletons. J.L. provided help on drawing schematic diagram. P.L. and C.G. wrote this manuscript. All authors discussed the results and com-mented on the manuscript.

## Competing interests

The authors declare no competing interests.
