## [Peer Review File · Nature Communications]

Bidirectionally promoting assembly order for ultrastiff and highly thermally conductive graphene fibresEditorial Note: This manuscript has been previously reviewed at another journal that is not operating a transparent peer review scheme. This document only contains reviewer comments and rebuttal letters for versions considered at *Nature Communications*.

REVIEWERS' COMMENTS

Reviewer #1 (Remarks to the Author):

This revised version tries to respond to reviewer's comments about the clarity of the formation mechanism of concentric structure and the novelty of research. The author's answers to those comments seem to be well treated and additional data the author added elucidate the mechanism of the inner fiber structure formation. Therefore, this manuscript meets high standards of Nature communication and it seems to be adequate to be published as is.

Response to the reviewer's comment

Reviewer #1

Comment: This revised version tries to respond to reviewer's comments about the clarity of the formation mechanism of concentric structure and the novelty of research. The author's answers to those comments seem to be well treated and additional data the author added elucidate the mechanism of the inner fiber structure formation. Therefore, this manuscript meets high standards of Nature communication and it seems to be adequate to be published as is.

Response: Thanks for your strong support and recognition. Your comments, as well as those from the other reviewer greatly improve this manuscript as compared with the original version. We benefit a lot from the peer-review process in most cases. While since graphene fibres and other 2D assembled fibres develop only for a few decades, some misunderstanding still persist in the eyes of some peers, which powers us to continue to create new knowledges on their solution quality, assembly process, structure controlling, and properties enhancement. We call upon that the whole community could also contribute to the development of 2D assembled fibers. Thanks for your comments again!